# Transmission of the atypical/Nor98 scrapie agent to Suffolk sheep with VRQ/ARQ, ARQ/ARQ, and ARQ/ARR genotypes

Eric D. Cassmann[1], Najiba Mammadova[1], S. Jo Moore[1], Sylvie Benestad[2], Justin J. Greenlee[1]*

1 Virus and Prion Research Unit, National Animal Disease Center, Agricultural Research Service, United States Department of Agriculture, Ames, Iowa, United States of America, 2 Norwegian Veterinary Institute, Oslo, Norway

* justin.greenlee@usda.gov

**Data Availability Statement:** All relevant data are within the manuscript and its Supporting Information files.

## Abstract

Scrapie is a transmissible spongiform encephalopathy that occurs in sheep. Atypical/Nor98 scrapie occurs in sheep that tend to be resistant to classical scrapie and it is thought to occur spontaneously. The purpose of this study was to test the transmission of the Atypical/Nor98 scrapie agent in three genotypes of Suffolk sheep and characterize the distribution of misfolded prion protein (PrP$^{Sc}$). Ten sheep were intracranially inoculated with brain homogenate from a sheep with Atypical/Nor98 scrapie. All sheep with the ARQ/ARQ and ARQ/ARR genotypes developed Atypical/Nor98 scrapie confirmed by immunohistochemistry, and one sheep with the VRQ/ARQ genotype had detectable PrP$^{Sc}$ consistent with Atypical/Nor98 scrapie at the experimental endpoint of 8 years. Sheep with mild early accumulations of PrP$^{Sc}$ in the cerebellum had concomitant retinal PrP$^{Sc}$. Accordingly, large amounts of retinal PrP$^{Sc}$ were identified in clinically affected sheep and sheep with dense accumulations of PrP$^{Sc}$ in the cerebellum.

## Introduction

Atypical/Nor98 scrapie (AS) is a fatal prion disease of sheep caused by a misfolded form of the prion protein. Unlike classical scrapie (CS), AS is thought to be a spontaneously occurring disease [1–3]. This is supported by the presence of AS in countries that are free of classical scrapie [4, 5]. It typically affects a single older sheep within a flock, and cases of AS are sporadic and isolated suggesting that natural transmission is unlikely.

The susceptibility of sheep to CS is closely related to polymorphisms in the prion protein gene (*PRNP*) [6, 7]. Polymorphisms associated with susceptibility or resistance to CS occur at codons 136, 154, and 171. Sheep with the $V_{136}R_{154}Q_{171}$ and $A_{136}R_{154}Q_{171}$ haplotypes are susceptible to CS; however, the amino acid polymorphisms $A_{136}$, $R_{154}$, and $R_{171}$ are associated with relative resistance [8–10]. Conversely, naturally occurring cases of AS arise in sheep with the AHQ, ARQ, and ARR haplotypes, and a polymorphism substituting phenylalanine (F) at codon 141 in the *PRNP* gene increases the risk of AS [11–13].

**Funding:** This research was funded in its entirety by congressionally appropriated funds to the United States Department of Agriculture, Agricultural Research Service. The funders of the work did not influence study design, data collection and analysis, decision to publish, or the preparation of the manuscript. The findings and conclusions in this publication are those of the author(s) and should not be construed to represent any official USDA or U.S. Government determination or policy. This research was supported in part by appointments (N. Mammadova and S. Jo Moore) to the Agricultural Research Service (ARS) Research Participation Program administered by the Oak Ridge Institute for Science and Education (ORISE) through an interagency agreement between the U.S. Department of Energy (DOE) and the U.S. Department of Agriculture (USDA). ORISE is managed by ORAU under DOE contract number DESC0014664. All opinions expressed in this paper are the author's and do not necessarily reflect the policies and views of USDA, ARS, DOE, or ORAU/ORISE.

**Competing interests:** The authors have declared that no competing interests exist.

Several experiments have demonstrated the ability of the AS agent to transmit within the natural host after intracranial inoculation [14–16]. One study found that the AS agent could transmit after a high oral dose of AS brain homogenate [17]. Nonetheless, AS is still considered unlikely to transmit under field conditions; therefore, eradication and surveillance programs for CS have allowed exceptions for AS. As research into AS unfolds, the biological relevance of this disease is gaining attention. Two studies have demonstrated phenotype changes in AS that imply a possible origin for classical scrapie [18] and classical BSE [19]. The present study was designed to generate AS brain material for subsequent projects to investigate interspecies transmission events. Herein, we report our findings after the experimental transmission of AS in sheep with the VRQ/ARQ, ARQ/ARQ, and ARQ/ARR genotypes. This study validates previous work on these genotypes and documents the early accumulation of PrP^Sc in the retina of sheep with AS.

## Results and discussion

All three genotypes of sheep, VRQ/ARQ, ARQ/ARQ, and ARQ/ARR, were susceptible to the AS agent after intracranial inoculation of donor brain homogenate. The diagnosis of AS was confirmed by enzyme immunoassay (EIA) and immunohistochemistry (IHC) with the latter being confirmative. Previous studies have demonstrated experimental transmission of AS to AHQ/AHQ [14, 15] and ARQ/ARQ [16] genotype sheep after intracerebral transmission. Another study showed a phenotypic shift from AS to CH1641-like classical scrapie in a sheep with the AHQ/AHQ genotype [18]. In this study, sheep with the ARQ/ARR genotype had the shortest incubation period ranging from 4.9 years to the experimental endpoint of 8 years (Table 1), and the attack rate was 100% (5/5). Clinical signs were observed in all ARQ/ARR sheep except for a single wether that was culled early to help establish experimental endpoints. Three ARQ/ARR genotype sheep were euthanized due to clinical neurologic disease 4.9–6.7 years post-inoculation. Out of the three genotypes examined, only the ARQ/ARR genotype sheep developed clinical neurologic disease within the eight-year incubation period. In clinically neurologic sheep, we observed stiff legged and hypermetric ataxia (dysmetria), abnormal

**Table 1. Results of atypical scrapie transmission in Suffolk sheep.**

| Animal no. | *PRNP* genotype | Years post-inoculation | Death status | Clinical signs | Retina | Cerebellum | |
|---|---|---|---|---|---|---|---|
| | | | | | IHC | IHC | EIA |
| 937 | ARQ/ARQ | 3.9* | Euthanized | No | + | + | + |
| 958 | ARQ/ARQ | 8.1 | End of study | No | + | + | + |
| 804 | ARQ/ARR | 6.7 | Euthanized | Yes | + | + | + |
| 927 | ARQ/ARR | 4.9 | Euthanized | Yes | + | + | + |
| 929 | ARQ/ARR | 8.1 | End of Study | Yes‡ | + | + | + |
| 933 | ARQ/ARR | 6.4† | Euthanized | No | + | + | + |
| 948 | ARQ/ARR | 6.7 | Euthanized | Yes | + | + | + |
| 926 | VRQ/ARQ | 1.2* | Found dead | No | neg | neg | neg |
| 943 | VRQ/ARQ | 8.1 | End of study | No | + | + | neg |
| 971 | VRQ/ARQ | 2.9* | Euthanized | No | neg | neg | neg |

A summary of results from three different genotypes of Suffolk sheep inoculated with atypical scrapie brain homogenate. Years post-inoculation indicates the incubation period in sheep with clinical disease or the survival times in sheep with early intercurrent disease (*). Sheep 933 was preliminarily culled to help determine an appropriate study endpoint since only a single sheep (937) was IHC positive up to that point and no clinical disease had been observed in any sheep yet (†). Sheep 929 was noted to have early mild non-specific clinical signs at the end of the study (‡). Abbreviations: *PRNP*, prion protein gene; IHC, immunohistochemistry; EIA, enzyme immunoassay.

rear stance, generalized tremors, tremors of the lips, weight loss, and generalized malaise. The spectrum of clinical signs was comparable to other reports of experimental AS in sheep [14, 15]. Three ARQ/ARR genotype sheep (804, 927 and 948) with the most severe dysmetria also had the greatest amount of cerebellar PrP^Sc. Since dysmetria is typical of animals with cerebellar disease [20], the tendency to observe this as the most consistent and severe neurologic sign is likely related to the characteristic cerebellar accumulation of PrP^Sc in sheep with AS. The ARQ/ARQ genotype had a long incubation period and remained clinically asymptomatic, as also reported by Okada et al. [16].

Sheep with the ARQ/ARQ genotype were positive for AS PrP^Sc by IHC (2/2). One positive wether remained asymptomatic and was necropsied at the experimental endpoint; whereas, the other sheep was culled due to intercurrent disease around four years post-inoculation. Out of the three original VRQ/ARQ genotype sheep, a single presymptomatic wether had PrP^Sc in the cerebellum and retina at the experimental endpoint of 8.1 years. The other two sheep succumbed to intercurrent disease, and they did not have detectable PrP^Sc by means of IHC or EIA at 1.2- and 2.9-years post-inoculation. The VRQ allele, that is generally associated with susceptibility to classical scrapie, is usually absent from naturally occurring AS cases [5, 12, 21]. However, in a study of AS cases from Great Britain, a single VRQ/ARQ case was reported [22]. The prolonged incubation period after intracranial inoculation of AS in a VRQ/ARQ genotype sheep is compatible with the low prevalence in field cases of AS. In fact, field cases of AS often have a polymorphism substituting phenylalanine (F) at codon 141 in the *PRNP* gene, and most cases have either the $AF_{141}RQ$ or AHQ alleles [12]. All of the sheep in this study contained the amino acid leucine (L) at codon 141.

In order to confirm that sheep had AS and rule out concomitant infection with classical scrapie, all tissues were examined by IHC for PrP^Sc. The distribution of PrP^Sc in the brains of sheep was consistent with AS. Immunolabeling of PrP^Sc appeared as granular and punctate deposits and was largely restricted to the molecular layer of the cerebellum (Fig 1A). Small amounts of punctate and granular staining were also seen in the cerebral cortex, basal nuclei, thalamus, and midbrain. In classical scrapie, PrP^Sc is found in the dorsal motor nucleus of the vagus nerve (DMNV), one of the early sites of central nervous system accumulation, and in the lymphoid tissue [3]. In the present experiment, PrP^Sc was observed in the spinal trigeminal tract (Fig 1B), and there was a lack of staining for PrP^Sc in the DMNV (Fig 1C). Additionally, no PrP^Sc was detectable by IHC in the lymphoid or peripheral tissues of any sheep; it remained confined to the CNS. Other studies have demonstrated infectivity in peripheral and lymphoid tissues that were IHC negative [17, 23]. This distribution of PrP^Sc in the present study was consistent with AS in sheep [1, 14]. Furthermore, all genotypes of sheep had similar PrP^Sc distributions; however, the density of staining was less severe in asymptomatic ARQ/ARQ and VRQ/ARQ genotype sheep. Given a longer incubation period culminating in clinical disease, it is expected that these genotypes would develop more severe PrP^Sc deposition similar to ARQ/ARR genotype sheep. PrP^Sc was also found in the spinal cords of each genotype of sheep. Staining of PrP^Sc appeared as small particulate or fine granular deposits in the dorsal horn. In sheep with the ARQ/ARR genotype, there was minimal PrP^Sc and it was usually observed in the cervical cord alone. Sheep 929 that lived to the experimental endpoint had PrP^Sc in both the cervical and thoracic cord segments. In sheep 958 (ARQ/ARQ) and 943 (VRQ/ARQ), there was a mild amount of PrP^Sc in the dorsal horn of the cervical, thoracic, and lumbar spinal cord segments. This differs from classical scrapie that involves the entire grey matter of the spinal cord in late stage disease [24].

We performed both IHC and EIA on cerebellum, cerebrum (parietal cortex), and medulla oblongata at the level of the obex. For the ARQ/ARR and ARQ/ARQ genotype sheep, there was 100% (7/7) agreement between IHC and EIA at detecting PrP^Sc in the cerebellum. In

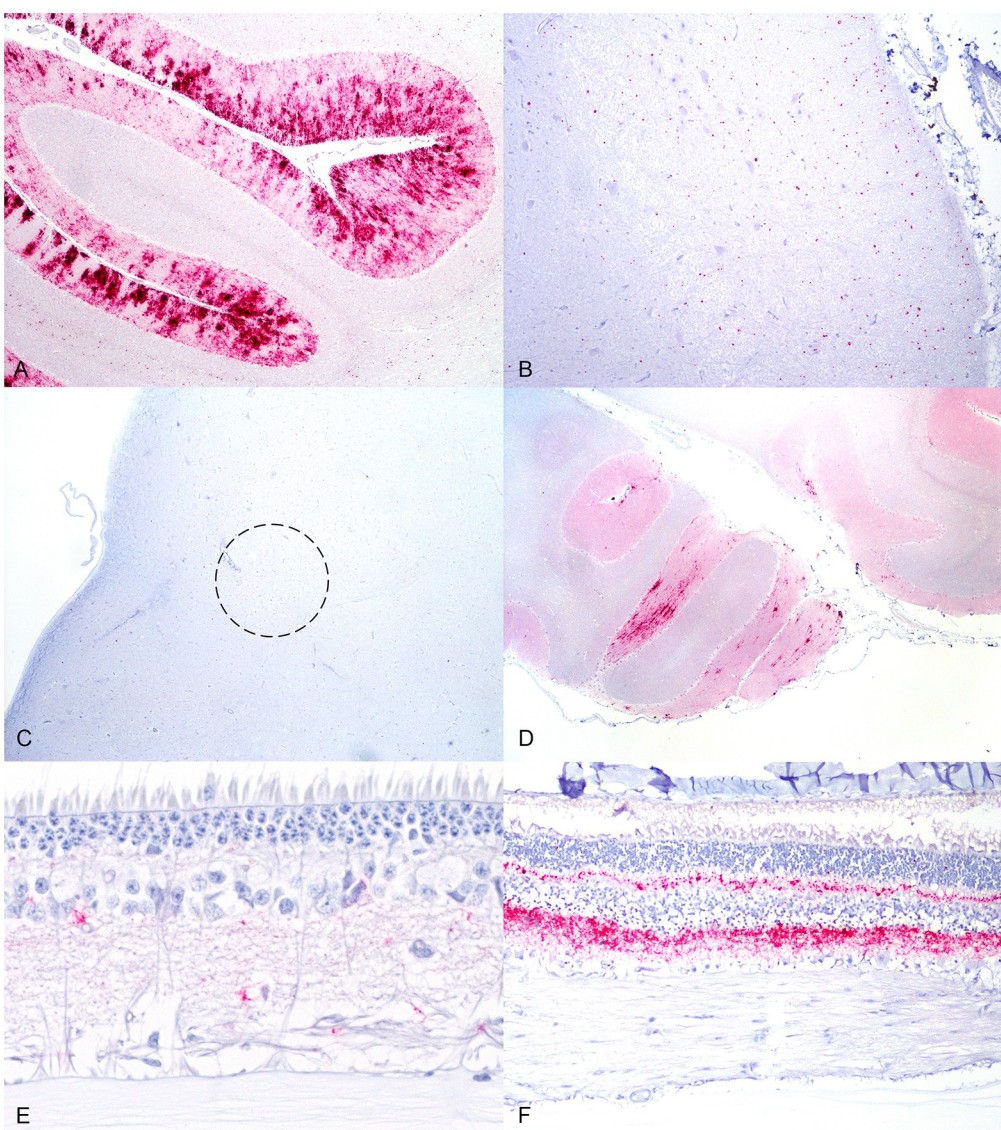

**Fig 1. Immunoreactivity of PrP$^{Sc}$ in sheep with atypical scrapie.** (A) There is a large amount of PrP$^{Sc}$ (red color) within the molecular layer of cerebellum in sheep 958 (ARQ/ARQ). (B) PrP$^{Sc}$ (red color) is confined to the spinal trigeminal tract in the medulla oblongata in sheep 948 (ARQ/ARR). (C) The dorsal motor nucleus of the vagus nerve (circle) is devoid of PrP$^{Sc}$ in sheep 948. (D) There are multifocal patchy aggregates of PrP$^{Sc}$ (red color) in the molecular layer of the cerebellum in sheep 943 (VRQ/ARQ). (E) A small amount of PrP$^{Sc}$ (red color) is present in the retina of sheep 943. (F) In sheep 958 there are large amounts of PrP$^{Sc}$ (red color) in the plexiform layers of the retina.

contrast, the single positive VRQ/ARQ sheep was IHC positive and EIA negative in the cerebellum. This discrepancy was presumably due to the patchy and sparse distribution of PrP$^{Sc}$ in the cerebellum (Fig 1D). PrP$^{Sc}$ was rarely observed in other brain regions of this animal; however, PrP$^{Sc}$ was detected in the retina with IHC (Fig 1E). Moreover, retinal PrP$^{Sc}$ was present in each genotype of sheep with atypical scrapie PrP$^{Sc}$ in the cerebellum. In clinical sheep with abundant cerebellar PrP$^{Sc}$, there were large amounts of PrP$^{Sc}$ in the retina (Fig 1F). PrP$^{Sc}$ occurred mostly in the inner and outer plexiform layers, but some minimal labeling was seen in the ganglion and nuclear cell layers. Other reports that describe atypical scrapie do not report retinal PrP$^{Sc}$ [1–5, 14–16, 25, 26]. In this study, sheep intracranially inoculated with the

atypical scrapie agent accumulated retinal PrP$^{Sc}$ in the early stages of disease concomitant with cerebellar PrP$^{Sc}$. This is significant because, sequentially, retinal PrP$^{Sc}$ accumulates early in disease; therefore, IHC of retinal tissue may be more sensitive compared to non-cerebellar brain regions.

This experiment demonstrated the transmission of atypical scrapie to three genotypes of sheep after intracranially inoculation, and it is the first study demonstrating experimental transmission to sheep with a VRQ/ARQ *PRNP* genotype. Additionally, atypical scrapie is further characterized by demonstrating early accumulation of PrP$^{Sc}$ in the retina of experimentally inoculated sheep.

## Materials and methods

Animals for this experiment were derived from a known scrapie-free flock at the United States Department of Agriculture National Animal Disease Center in Ames, IA. This study used ten Suffolk sheep, nine wethers and one ewe. Nine sheep were 1 year old at the time of inoculation. A single sheep, #958, was 2 years old. Sheep in this study had three distinct *PRNP* genotypes: ARQ/ARQ, ARQ/ARR, and VRQ/ARQ. The genotypes were determined using polymerase chain reaction and Sanger sequencing as previously described [27]. Sheep were homozygous at other known polymorphic sites M112, G127, M137, S138, L141, R151, M157, N176, H180, Q189, T195, T196, R211, Q220, and R223.

The inoculum for this experiment was cerebral homogenate from an AHQ/ARH genotype sheep with atypical scrapie from Norway (Hedalen). The inoculum was obtained through a collaboration with Sylvie Benestad at the Norwegian Veterinary Institute. The brain homogenate was prepared as a 10% w/v homogenate. Sheep were intracranially inoculated with 1 ml (0.1 grams) of brain homogenate. The procedure has been described previously [28]. Briefly, the sheep were anesthetized with xylazine and a surgical field was prepped over the junction of parietal and frontal bones. A 1-cm skin incision was made, and then a 1-mm hole was drilled along the midline of the calvaria. A 9-cm spinal needle was inserted through the hole, and the inoculum was injected into the cranium. Sheep were kept in a biosecurity level 2 indoor pen for two weeks following inoculation and then moved to an outdoor area. They were fed a daily ration of pelleted and loose alfalfa hay. Sheep were monitored daily for any maladies or other clinical signs consistent with scrapie. The experimental endpoint for this experiment included the earliest of either unequivocal neurologic disease or 8 years post-inoculation. The final 8-year endpoint was established by performing a preliminarily cull of sheep 933 to help determine an appropriate endpoint. Sheep were euthanized at the onset of clinical disease or untreatable intercurrent disease. The method of euthanasia was intravenous administration of sodium pentobarbital as per label directions or as directed by an animal resources attending veterinarian. Clinical signs of disease included abnormalities in gate and/or stance, and ataxia.

A full post-mortem examination was performed on each sheep, and a routine set of tissues were collected consistent with previous experiments [29, 30]. A duplicate set of the following tissues were frozen or saved to 10% buffered neutral formalin: brain, spinal cord, pituitary, trigeminal ganglia, eyes, sciatic nerve, third eyelid, palatine tonsil, pharyngeal tonsil, lymph nodes (mesenteric, retropharyngeal, prescapular, and popliteal), spleen, esophagus, forestomaches, intestines, rectal mucosa, thymus, liver, kidney, urinary bladder, pancreas, salivary gland, thyroid gland, adrenal gland, trachea, lung, turbinate, nasal planum, heart, tongue, masseter, diaphragm, triceps brachii, biceps femoris, and psoas major. Formalin fixed tissues were processed, paraffin embedded, and sectioned at optimal thickness (brain, 4 μm; lymphoid, 3 μm; and other, 5 μm) for hematoxylin and eosin staining and IHC. For IHC, a cocktail of the monoclonal anti-PrP$^{Sc}$ antibodies F89/160.1.5 [31] and F99/97.6.1 [32] was applied at a

concentration of 5 μg/mL using an automated stainer. Frozen portions of cerebellum, parietal cerebral cortex, and brainstem at the level of the obex were homogenized and tested for the presence of PrP<sup>Sc</sup> using a commercially available EIA (HerdChek; IDEXX Laboratories, Westbrook, ME) according to kit instructions.

## Ethics statement

The laboratory and animal experiments were conducted in Biosafety Level 2 spaces that were inspected and approved for importing prion agents by the US Department of Agriculture, Animal and Plant Health Inspection Service, Veterinary Services. The studies were done in accordance with the Guide for the Care and Use of Laboratory Animals (Institute of Laboratory Animal Resources, National Academy of Sciences, Washington, DC, USA) and the Guide for the Care and Use of Agricultural Animals in Research and Teaching (Federation of Animal Science Societies, Champaign, IL, USA). The protocols were approved by the Institutional Animal Care and Use Committee at the National Animal Disease Center (protocol numbers: 3908 and ARS-2777), which require species-specific training in animal care for all staff handling animals.

## Acknowledgments

The authors wish to thank Rylie Frese, Kevin Hassall, Joe Lesan, Leisa Mandell, and Trudy Tatum for excellent technical support. The findings and conclusions in this publication are those of the author(s) and should not be construed to represent any official USDA or U.S. Government determination or policy. Mention of trade names or commercial products in this article is solely for the purpose of providing specific information and does not imply recommendation or endorsement by the Department of Agriculture. The Department of Agriculture is an equal-opportunity provider and employer.

## Author Contributions

**Conceptualization:** Sylvie Benestad, Justin J. Greenlee.

**Data curation:** Eric D. Cassmann, Najiba Mammadova.

**Formal analysis:** Eric D. Cassmann, Najiba Mammadova, S. Jo Moore.

**Investigation:** Eric D. Cassmann, Najiba Mammadova, S. Jo Moore, Sylvie Benestad, Justin J. Greenlee.

**Methodology:** Justin J. Greenlee.

**Project administration:** Justin J. Greenlee.

**Resources:** Sylvie Benestad, Justin J. Greenlee.

**Supervision:** Sylvie Benestad, Justin J. Greenlee.

**Writing – original draft:** Eric D. Cassmann, Najiba Mammadova.

**Writing – review & editing:** S. Jo Moore, Sylvie Benestad, Justin J. Greenlee.

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
