## [Decision Letter · Decision Letter 0]

18 Dec 2020

PONE-D-20-36348

Transmission of the atypical/Nor98 scrapie agent to Suffolk sheep with VRQ/ARQ, ARQ/ARQ, and ARQ/ARR genotypes

PLOS ONE

Dear Dr. Justin Greenlee,  

Thank you for submitting your manuscript to PLOS ONE. After careful consideration, we feel that it has merit but does not fully meet PLOS ONE’s publication criteria as it currently stands. Therefore, we invite you to submit a revised version of the manuscript that addresses the points raised during the review process.

The reviewers raised several and important criticisms which need to be addressed. I think that following their indications the study will be consistently improved. Please, consider also the quality of the immunohistochemical pictures which need to be . 

We look forward to receiving your revised manuscript.

Kind regards,

Gianluigi Zanusso

Academic Editor

PLOS ONE

Journal Requirements:

"This research was funded in its entirety by congressionally appropriated funds to the United States

 Department of Agriculture, Agricultural Research Service. The funders of the work did not

influence study design, data collection and analysis, decision to publish, or the preparation of the

manuscript.

This research was supported in part by an appointment to the Agricultural Research Service

(ARS) Research Participation Program administered by the Oak Ridge Institute for Science and

Education (ORISE) through an interagency agreement between the U.S. Department of Energy

(DOE) and the U.S. Department of Agriculture (USDA). ORISE is managed by ORAU under DOE

contract number DE-SC0014664. All opinions expressed in this paper are the author's and do not

necessarily reflect the policies and views of USDA, ARS, DOE, or ORAU/ORISE."

"This research was funded in its entirety by congressionally appropriated funds to the United States Department of Agriculture, Agricultural Research Service. The funders of the work did not influence study design, data collection and analysis, decision to publish, or the preparation of the manuscript."

Reviewers' comments:

Reviewer's Responses to Questions

**Comments to the Author**

1. Is the manuscript technically sound, and do the data support the conclusions?

Reviewer #1: Partly

Reviewer #2: Yes

Reviewer #3: Yes

2. Has the statistical analysis been performed appropriately and rigorously? 

Reviewer #1: N/A

Reviewer #2: N/A

Reviewer #3: N/A

3. Have the authors made all data underlying the findings in their manuscript fully available?

Reviewer #1: Yes

Reviewer #2: Yes

Reviewer #3: Yes

4. Is the manuscript presented in an intelligible fashion and written in standard English?

Reviewer #1: Yes

Reviewer #2: Yes

Reviewer #3: No

5. Review Comments to the Author

Reviewer #1: This manuscript describes the outcome in ten sheep of three different prion protein genotypes after intracranial inoculation with atypical scrapie homogenate from a single source. The authors describe clinical presentation (if any), survival times and pathological results in a range of tissues collected. The study was carried out to provide brain material for subsequent interspecies transmissions (so the lack of controls is excusable) and is thus not a typical research article exploring a hypothesis but provides nevertheless useful information worth publishing. The study confirms previous work and is the first to show transmission to ARQ/VRQ sheep. In addition, the study demonstrates early accumulation of PrPSc in the retina.

As this study does not test a hypothesis the question arises why the authors used sheep with prion protein genotypes where the outcome was not known at all. Surely, if tissue generation was the main purpose one would have selected sheep with genotypes with known susceptibility. Overall, there are some issues that I feel require further clarification, which I outline below:

1) What is the significance of early accumulation of PrPSc in the retina? Is it to suggest that there may be ERG changes that may be detectable in the live animal? Could it be used as a simple test in dead animals where aqueous humor could be collected post-mortem and checked for PrPSc? This may be worth discussing.

2) What was the reasoning for the sample size? Why selecting two ARQ/ARQ sheep, five ARQ/ARR sheep and two VRQ/ARQ sheep?

3) What was the rationale for selecting a single brain as inoculum from a sheep that had an AHQ/ARH genotype, basically none of the haplotypes matching those of the recipient sheep? The only experimental study where heterologous donor and recipient genotypes were used (Simmons et al.; reference 18) demonstrated a potential shift of the prion strain, so would it not be ‘safer’ to use homologous donor and recipient genotypes for atypical scrapie tissue generation? The effect of different genotypes between donor and recipient on survival time and phenotype is certainly known for classical scrapie so I would have expected at least some discussion about this. An alternative study design could have included a group of sheep that had a least some matched haplotype, e.g. AHQ or ARH.

In addition, please consider the following:

Line 26. An ‘a’ is missing after ‘is’.

Line 59. The early cull is not mentioned in M&M.

Table 1. This table is too wide to fit on the page so I cannot read all the columns. Is there a column for EIA results? In general, I prefer survival times displayed in either months or even days because it is difficult to interpret decimals in years.

Lines 153+. More detail is needed to comply at least with the ARRIVE guidelines. Sex and age are not mentioned; the use of a wether is mentioned in line 58 – were all sheep of the same sex? It is common to refer to previous publications when describing experimental procedures but I would prefer more detail in this manuscript: more detail about donor brain (from fallen stock? Active surveillance? Was it heat treated or treated with antibiotics?). I am slightly concerned about the pharmaceutical treatment of the sheep; xylazine causes sedation rather than anaesthesia and has limited analgesic properties, which is dose-dependent, so stating the dose would be beneficial. I would have assumed that area preparation and drilling a hole would require some anaesthesia, even if it is just local. There is no indication of analgesia, which I found astonishing, because the use of opiates or NSAID is now common for surgical procedures. The study was reviewed by an ethics committee (to my surprise the authors entered N/A in the Ethics statement in the submission form) and I would have assumed that analgesia is used – please provide more detail.

Lastly, I found that the quality of the photos could be improved, particularly c, but it may be the resolution in the pdf file.

Reviewer #2: Dear Authors,

Your manuscript, "Transmission of the atypical/Nor98 scrapie agent to Suffolk sheep with VRQ/ARQ,

ARQ/ARQ, and ARQ/ARR genotypes" provides additional data regarding susceptibility of sheep with known genotypes to AS. The description of retinal involvement of AS is novel.

A few comments to consider:

1. Addition of the susceptibility status of ARQ sheep to CS near line 34 would help provide a more complete history.

2. Line 53, insert the words "of AS" after "transmission" and before "to"

3. Table 1 is cutoff in the PDF version provided by PLoS One. I could not view all the data. I am not sure if this is an author or journal oversite. Unfortunately a table only file was not available.

4. Lines 87-89 should be included in the introduction after line 37 to help set-up the study. Similar information could be provided as needed again in 87.

5. In the VRQ group the authors claim only 1 of 3 sheep developed AS based on testing. It is almost unfair to include the two negative sheep in this group, as they left the study very early on. This is discussed, but perhaps they should be excluded from the ratio used in the abstract?

Reviewer #3: In this manuscript the Authors describe the transmission of atypical scrapie to Suffolk sheep with different genotypes. The manuscript can be of interest for scientists working on prions and can be published on PLOS One. However, before publication, the text should be amended with minor revisions to make it more clear in some points.

Here the requested changes:

- Please move lines 67-71 to line 65, after the sentence citing the references 14 and 15.

- Please move the sentence at lines 65-66 to lines 70-71 and rewrite the sentence as follows: The ARQ/ARQ genotype had a long incubation period and remained clinically asymptomatic, as also reported by Okada et al.

- Delete "In the presente study"

- Line 68: I think that the term degree is not appropriate.

- Line 83 please write "Out of the three"

- Line 84 please write correctly the term wether

- Lines 91-92: this sentence is more appropriate in the introduction

- Line 93: this sentence is more appropriate when animals are described

- Lines 108-109: I do not understand the meaning of this sentence...is it an hypothesis or a statement? I do not think it can be a statement

- Lines 140-141: I think that it is more correct to say....demonstrating the possibility of an early accumulation of PrPsc in the retina.....it is an experimental challenge via i.c. route thus it cannot be excluded that this could have influenced the deposition in the retina

- Line 160: I never heard the term maladies in English, but I am not an English mother tongue.

6. PLOS authors have the option to publish the peer review history of their article (what does this mean?). If published, this will include your full peer review and any attached files.

Reviewer #1: No

Reviewer #2: No

Reviewer #3: No

---

## [Author Response · Author response to Decision Letter 0]

7 Jan 2021

Our responses also are available in a separate file with formatting that makes it easier to distinguish comments from responses.

Atypical Scrapie_PONE-20-45383_Response to reviewers

Journal Requirements:

The following changes were made on our resubmission: the title was centered, and the headings were changed to sentence case. 

"This research was funded in its entirety by congressionally appropriated funds to the United States Department of Agriculture, Agricultural Research Service. The funders of the work did not influence study design, data collection and analysis, decision to publish, or the preparation of the manuscript.

This research was supported in part by an appointment to the Agricultural Research Service (ARS) Research Participation Program administered by the Oak Ridge Institute for Science and Education (ORISE) through an interagency agreement between the U.S. Department of Energy (DOE) and the U.S. Department of Agriculture (USDA). ORISE is managed by ORAU under DOE contract number DE-SC0014664. All opinions expressed in this paper are the author's and do not necessarily reflect the policies and views of USDA, ARS, DOE, or ORAU/ORISE."

"This research was funded in its entirety by congressionally appropriated funds to the United States Department of Agriculture, Agricultural Research Service. The funders of the work did not influence study design, data collection and analysis, decision to publish, or the preparation of the manuscript."

This research was funded in its entirety by congressionally appropriated funds to the United States Department of Agriculture, Agricultural Research Service. The funders of the work did not influence study design, data collection and analysis, decision to publish, or the preparation of the manuscript. The findings and conclusions in this publication are those of the author(s) and should not be construed to represent any official USDA or U.S. Government determination or policy.

This research was supported in part by appointments (N. Mammadova and S. Jo Moore) to the Agricultural Research Service (ARS) Research Participation Program administered by the Oak Ridge Institute for Science and Education (ORISE) through an interagency agreement between the U.S. Department of Energy (DOE) and the U.S. Department of Agriculture (USDA). ORISE is managed by ORAU under DOE contract number DE-SC0014664. All opinions expressed in this paper are the author's and do not necessarily reflect the policies and views of USDA, ARS, DOE, or ORAU/ORISE.

The ethics statement is included as a subheading in the Materials and methods section. 

Reviewer #1: This manuscript describes the outcome in ten sheep of three different prion protein genotypes after intracranial inoculation with atypical scrapie homogenate from a single source. The authors describe clinical presentation (if any), survival times and pathological results in a range of tissues collected. The study was carried out to provide brain material for subsequent interspecies transmissions (so the lack of controls is excusable) and is thus not a typical research article exploring a hypothesis but provides nevertheless useful information worth publishing. The study confirms previous work and is the first to show transmission to ARQ/VRQ sheep. In addition, the study demonstrates early accumulation of PrPSc in the retina. As this study does not test a hypothesis the question arises why the authors used sheep with prion protein genotypes where the outcome was not known at all. Surely, if tissue generation was the main purpose one would have selected sheep with genotypes with known susceptibility. Overall, there are some issues that I feel require further clarification, which I outline below:

1) What is the significance of early accumulation of PrPSc in the retina? Is it to suggest that there may be ERG changes that may be detectable in the live animal? Could it be used as a simple test in dead animals where aqueous humor could be collected post-mortem and checked for PrPSc? This may be worth discussing.

Thank you for your input. In order to clarify the significance of early retinal accumulation, added the following text: 

“This is significant because, sequentially, retinal PrPSc accumulates early in disease; therefore, IHC of retinal tissue may be more sensitive compared to non-cerebellar brain regions.”

2) What was the reasoning for the sample size? Why selecting two ARQ/ARQ sheep, five ARQ/ARR sheep and two VRQ/ARQ sheep?

These were the sheep that were available from our scrapie free breeding herd the year the experiment commenced. 

3) What was the rationale for selecting a single brain as inoculum from a sheep that had an AHQ/ARH genotype, basically none of the haplotypes matching those of the recipient sheep? 

This is the inoculum that was provided by collaborators in Norway. The collaboration (import of samples) began prior to atypical scrapie being described in the US.

This was the only inoculum we had with L141 that matched our recipient sheep. 

The only experimental study where heterologous donor and recipient genotypes were used (Simmons et al.; reference 18) demonstrated a potential shift of the prion strain, so would it not be ‘safer’ to use homologous donor and recipient genotypes for atypical scrapie tissue generation? 

It seems it would be safer if we wanted to “guarantee” transmission, however, a homologous genotype recipient was not available in our scrapie free flock. 

A paper from Loiacono et al in 2009 described the diagnostic results from 6 sheep in the US with atypical scrapie. None of the six sheep had the ARH haplotype and only one had the AHQ haplotype, so it also seems reasonable to study atypical scrapie in sheep with haplotypes that were represented in the first 6 US cases (ARQ and ARR). Note: both FL and LL 141 were represented in these six sheep as well. 

The effect of different genotypes between donor and recipient on survival time and phenotype is certainly known for classical scrapie so I would have expected at least some discussion about this. An alternative study design could have included a group of sheep that had a least some matched haplotype, e.g. AHQ or ARH.

Matched haplotype sheep were not available for use from our scrapie free breeding flock.

In addition, please consider the following:

Line 26. An ‘a’ is missing after ‘is’.

Thank you for pointing this out. The suggestion was incorporated. 

Line 59. The early cull is not mentioned in M&M.

Thank you for your input. We incorporated the following text in our methods section. 

The methods now read “The experimental endpoint for this experiment included the earliest of either unequivocal neurologic disease or 8 years post-inoculation. The final 8-year endpoint was established by performing a preliminarily cull of sheep 933 to help determine an appropriate endpoint.”

Table 1. This table is too wide to fit on the page so I cannot read all the columns. Is there a column for EIA results? In general, I prefer survival times displayed in either months or even days because it is difficult to interpret decimals in years.

Yes, there are EIA results. 

We regret that you were unable to see the whole table. We are not sure how to rectify this within the manuscript, since we followed the author guidelines to not force tables within margins. We have submitted an additional file with the contents of this table specifically to make it easier for reviewers to view. 

Lines 153+. More detail is needed to comply at least with the ARRIVE guidelines. Sex and age are not mentioned; the use of a wether is mentioned in line 58 – were all sheep of the same sex? It is common to refer to previous publications when describing experimental procedures but I would prefer more detail in this manuscript: more detail about donor brain (from fallen stock? Active surveillance? Was it heat treated or treated with antibiotics?). 

Thank you for pointing out that we neglected to include the information about sex and age. We have modified our methods section to include sex and age information in accordance with ARRIVE. 

I am slightly concerned about the pharmaceutical treatment of the sheep; xylazine causes sedation rather than anesthesia and has limited analgesic properties, which is dose-dependent, so stating the dose would be beneficial. I would have assumed that area preparation and drilling a hole would require some anesthesia, even if it is just local. There is no indication of analgesia, which I found astonishing, because the use of opiates or NSAID is now common for surgical procedures. 

This protocol was approved over ten years ago when these sheep were inoculated. Our current procedure includes the use of Banamine. We avoid opioids in sheep. Opioid agonists have less analgesic effects in sheep and can cause excitation and behavioral changes (Kastner, 2006). 

The most painful part of the procedure is the small skin incision. Some innervation of the periosteum may also result in minimal to mild bone pain. We now provide some Banamine, but it should be noted that neither now nor before did any sheep display behaviors consistent with post-inoculation discomfort (head shaking, pressing, rubbing, decreased appetite). Therefore, the previous protocol seemed to adequately provide analgesia even with the analgesic properties of Xylazine alone. In sheep, xylazine has been shown to have significant antinociceptive effects to painful electrical stimuli even at low doses that have minimal sedative effects (Grant, 2004). The sedative effectives of xylazine are dose dependent. 

“Use of xylazine as a sole agent can be a practical method of inducing recumbency for completion of nonpainful procedures. (Reza, 2016)” At the level of sedation we use, sheep do not respond to any stimulus from the procedure during inoculation; therefore, the concurrent use of a dissociative anesthetic like ketamine seems unnecessary. Inoculation is a short procedure, so the incorporation of ketamine may actually increase the chances for side effects by prolonging the sedation. Sheep are known to be susceptible to pulmonary effects from xylazine. Furthermore, use of ketamine for neuroanesthesia is debated due to the possibility of deleterious effects on cerebral circulation. 

Kästner, S. B. A2‐agonists in sheep: a review. Vet Anaesth Analg 33, 79–96 (2006). 

GRANT, C. & UPTON, R. Comparison of the analgesic effects of xylazine in sheep via three different administration routes. Aust Vet J 82, 304–307 (2004). 

Reza Seddighi, Thomas J. Doherty, Field Sedation and Anesthesia of Ruminants, Veterinary Clinics of North America: Food Animal Practice, Volume 32, Issue 3, 2016, Pages 553-570.

The study was reviewed by an ethics committee (to my surprise the authors entered N/A in the Ethics statement in the submission form) and I would have assumed that analgesia is used – please provide more detail.

Our ethics statement was provided in the manuscript under the sub-heading “ethics statement”, and the study was approved by an ethics committee. The committee and professional evaluation of multiple veterinarians concluded that the analgesic effects of xylazine were sufficient for the pain elicited by a 1 cm skin incision and a 1 mm hole in skull. Again, currently we are also using Banamine, but that was not the case 10 years ago when these sheep were inoculated. 

Lastly, I found that the quality of the photos could be improved, particularly c, but it may be the resolution in the pdf file.

Per the editorial manager instructions:

All PDF files are created at 72 dpi resolution to make the file size manageable for downloading. Each image in the PDF file link to a higher resolution image is available for each image in the PDF file.

Reviewer #2: Dear Authors,

Your manuscript, "Transmission of the atypical/Nor98 scrapie agent to Suffolk sheep with VRQ/ARQ, ARQ/ARQ, and ARQ/ARR genotypes" provides additional data regarding susceptibility of sheep with known genotypes to AS. The description of retinal involvement of AS is novel.

A few comments to consider:

1. Addition of the susceptibility status of ARQ sheep to CS near line 34 would help provide a more complete history.

Thank you for this comment. We have incorporated the suggested change.

2. Line 53, insert the words "of AS" after "transmission" and before "to"

Thank you for this comment. We incorporated the suggested change.

3. Table 1 is cutoff in the PDF version provided by PLoS One. I could not view all the data. I am not sure if this is an author or journal oversite. Unfortunately a table only file was not available.

We regret that you were unable to see the whole table. We are not sure how to rectify this within the manuscript, since we followed the author guidelines to not force tables within margins. We have submitted this as an individual file for the reviewers.

4. Lines 87-89 should be included in the introduction after line 37 to help set-up the study. Similar information could be provided as needed again in 87.

Thank you for your suggestion. In the introduction, we set up the study by noting a difference in the genetic susceptibility of sheep to atypical scrapie by indicating what genotypes are typically represented (AHQ, ARQ, ARR) – new Lines 39-41. The rarity of VRQ genotype sheep with atypical scrapie is noted in the discussion. 

5. In the VRQ group the authors claim only 1 of 3 sheep developed AS based on testing. It is almost unfair to include the two negative sheep in this group, as they left the study very early on. This is discussed, but perhaps they should be excluded from the ratio used in the abstract?

Thank you for pointing this out. We agree and have changed the abstract accordingly. 

Reviewer #3: In this manuscript the Authors describe the transmission of atypical scrapie to Suffolk sheep with different genotypes. The manuscript can be of interest for scientists working on prions and can be published on PLOS One. However, before publication, the text should be amended with minor revisions to make it more clear in some points.

Here the requested changes:

- Please move lines 67-71 to line 65, after the sentence citing the references 14 and 15.

- Please move the sentence at lines 65-66 to lines 70-71 and rewrite the sentence as follows: The ARQ/ARQ genotype had a long incubation period and remained clinically asymptomatic, as also reported by Okada et al.

- Delete "In the presente study"

- Line 68: I think that the term degree is not appropriate.

Thank you for your comments and suggestions. We have incorporated them in the revised manuscript. 

- Line 83 please write "Out of the three"

Completed as suggested. 

- Line 84 please write correctly the term wether

Thank you for catching this. We have added this word to the MS Word dictionary, so it doesn’t get autocorrected again. 

- Lines 91-92: this sentence is more appropriate in the introduction

Thank you for your comments. We think that this information as written in the discussion is appropriate. In the introduction, we already discussed the increased risk for atypical scrapie associated with F141. In both locations, the information is technically/scientifically sound. No changes were made. 

- Line 93: this sentence is more appropriate when animals are described

This information is included in the methods section already. The discussion reiterates this information for the purposes of discussing the lack of F141 sheep in this study. It seems like a relevant discussion point to highlight this. No changes were made. 

- Lines 108-109: I do not understand the meaning of this sentence...is it an hypothesis or a statement? I do not think it can be a statement

This is consistent with a hypothesis. We simply are discussing what we expect would happen if the incubation period was allowed to continue in sheep with ARQ/ARQ and VRQ/ARQ genotypes.

- Lines 140-141: I think that it is more correct to say....demonstrating the possibility of an early accumulation of PrPsc in the retina.....it is an experimental challenge via i.c. route thus it cannot be excluded that this could have influenced the deposition in the retina

Thanks for your comment. We have made changes to this paragraph to address your concerns. 

- Line 160: I never heard the term maladies in English, but I am not an English mother tongue.

No changes were made.

6. PLOS authors have the option to publish the peer review history of their article (what does this mean?). If published, this will include your full peer review and any attached files.

Do you want your identity to be public for this peer review? For information about this choice, including consent withdrawal, please see our Privacy Policy.

Reviewer #1: No

Reviewer #2: No

Reviewer #3: No

---

## [Editor Report · Decision Letter 1]

21 Jan 2021

Transmission of the atypical/Nor98 scrapie agent to Suffolk sheep with VRQ/ARQ, ARQ/ARQ, and ARQ/ARR genotypes

PONE-D-20-36348R1

Dear Dr. Greenlee,

We’re pleased to inform you that your manuscript has been judged scientifically suitable for publication and will be formally accepted for publication once it meets all outstanding technical requirements.

Kind regards,

Gianluigi Zanusso

Academic Editor

PLOS ONE

---

## [Editor Report · Acceptance letter]

29 Jan 2021

PONE-D-20-36348R1 

Transmission of the atypical/Nor98 scrapie agent to Suffolk sheep with VRQ/ARQ, ARQ/ARQ, and ARQ/ARR genotypes 

Dear Dr. Greenlee:

I'm pleased to inform you that your manuscript has been deemed suitable for publication in PLOS ONE. Congratulations! Your manuscript is now with our production department. 

Kind regards, 

on behalf of

Dr. Gianluigi Zanusso 

Academic Editor

PLOS ONE